# Deep Learning Applications in Pancreatic Cancer

**DOI:** 10.3390/cancers16020436

**Published:** 2024-01-19

**Authors:** Hardik Patel, Theodoros Zanos, D. Brock Hewitt

**Affiliations:** 1Northwell Health—The Feinstein Institutes for Medical Research, Manhasset, NY 11030, USA; tzanos@northwell.edu; 2Department of Surgery, NYU Grossman School of Medicine, New York, NY 10016, USA; brock.hewitt@nyulangone.org

**Keywords:** artificial intelligence, deep learning, pancreatic cancer

## Abstract

**Simple Summary:**

The application of artificial intelligence (AI) in healthcare is expanding rapidly and is everchanging. This review focuses on the application of AI, specifically Deep Learning (DL), in the diagnosis, management, and monitoring of patients diagnosed with pancreatic cancer (PC), one of the most lethal gastrointestinal malignancies. A comprehensive review of DL modalities and applications through the difference stages in management of PC patients is discussed. From diagnostic and prognostic imaging, to postoperative care and utilization of AI to develop novel biomarkers. An emphasis is placed on reviewing the breadth of applications and methodologies that are revolutionizing the care of pancreatic cancer patients.

**Abstract:**

Pancreatic cancer is one of the most lethal gastrointestinal malignancies. Despite advances in cross-sectional imaging, chemotherapy, radiation therapy, and surgical techniques, the 5-year overall survival is only 12%. With the advent and rapid adoption of AI across all industries, we present a review of applications of DL in the care of patients diagnosed with PC. A review of different DL techniques with applications across diagnosis, management, and monitoring is presented across the different pathological subtypes of pancreatic cancer. This systematic review highlights AI as an emerging technology in the care of patients with pancreatic cancer.

## 1. Introduction

Pancreatic cancer (PC) is the fourth leading cause of cancer-related mortality in the United States and, if current trends continue, will become the second most fatal cancer by 2030 [1]. PC is transforming into a worldwide crisis with an estimated incidence of 355,317 cases by 2040 [2,3]. Over 90% of PC develops from cells of the exocrine system, such as pancreatic ductal adenocarcinoma (PDAC) and pancreatic acinar cell carcinoma, while the remainder of pancreatic tumors originate from endocrine cells, called neuroendocrine tumors, or pancreatic cystic lesions (PCLs), including intraductal papillary mucinous neoplasm (IPMN) and mucinous cystic neoplasms (MCNs). Some PCLs carry a risk of malignant transformation and an opportunity for curative resection prior to malignant transformation if detected early [3,4]. Early-stage PC is often indolent, presenting with vague abdominal symptoms, jaundice, weight loss, or no symptoms at all. As a result, only 10–15% of patients have a resectable disease at diagnosis while approximately 50% present with a metastatic disease. Although a biopsy is not needed prior to surgical resection when clinical suspicion is high, standard staging workup for PC involves a multiphasic cross-sectional imaging of the chest, abdomen, and pelvis with thin cuts through the pancreas. Furthermore, endoscopic interventions including esophagogastroduodenoscopy (EGD) with an endoscopic ultrasound (EUS) and fine-needle biopsy (FNB) or endoscopic retrograde cholangiopancreatography (ERCP) may be needed prior to the initiation of neoadjuvant chemotherapy when a tissue diagnosis is required or when the patient needs relief from biliary obstruction [5,6]. Finally, biomarkers such as CA 19-9 and CEA provide prognostic information and assist with clinical decision making. Despite multimodal treatment approaches with multi-agent chemotherapy, radiation, and surgical resection, the five-year survival rate of patients diagnosed with PC is 12%, making PC one of the most fatal of all gastrointestinal malignancies.

Over the past few years, artificial intelligence (AI) utilization has dramatically expanded across healthcare. AI is expected to revolutionize the healthcare industry by improving diagnostic and early disease detection accuracy, treatment deliveries, patient engagement and adherence, and streamlining administrative processes. A review published by Berbis et al. shows an exponential increase in the number of published research studies in the field of AI on the pancreas [7] (Figure 1).

AI-derived algorithms can be subdivided into supervised learning algorithms, which are trained using labeled data, and unsupervised learning algorithms, which are able to detect patterns in unstructured data without human input. A specific type of AI is machine learning (ML) which encompasses a set of both supervised techniques, such as support vector machines (SVMs) and random forest (RF), and unsupervised learning algorithms such as clustering. Radiomics is an emerging concept in the field of radiology and describes the process of feature extraction from clinical imaging (e.g., pixel intensity, shape, and texture) followed by a quantitative analysis, often with ML algorithms, to either classify or predict an outcome. A subset of ML, deep learning (DL) seeks to imitate the behavior of human intelligence through the application of large networks consisting of perceptrons, analogues to the human neuron (Figure 2).

A neural network (NN) is a network of perceptrons that takes a set of input variables with associated weights and formulates an output. Multiple layers can be interconnected in numerous ways to find a set of weights that can predict an output with the least amount of error and, in effect, predict an outcome. The goal of this article is to review the different applications and methods of DL and radiomics in the diagnosis and management of PC, specifically PDACs, PNETs, and PCLs.

## 2. Materials and Methods

An analysis of all English language publications from January 2019 to November 2023 was searched by accessing the PubMed database. The keywords utilized included “pancreatic cancer”, “deep learning”, “radiomics”, “large language models”, and “generative adversarial networks”. A total of 54 results were obtained including review articles, retrospective studies, and prospective studies. After the exclusion of repetitive and irrelevant content, a total of 26 studies were included in this review (Table 1). The article selection algorithm can be seen in Figure 3, including the specific DL/ML algorithm utilized in each study.

## 3. Results

### 3.1. Pancreatic Ductal Adenocarcinoma

#### 3.1.1. Applications in the Diagnosis of Pancreatic Ductal Adenocarcinoma

Patients diagnosed with PC are generally classified into four categories, resectable, borderline resectable, locally advanced, or metastatic, based on the results of cross-sectional imaging. In current clinical practice, CT is the most cost-effective and extensively used cross-sectional imaging modality to evaluate patients with or for PC [10]. However, the diagnosis of PC from CT remains a difficult task for radiologists due to high heterogeneity among suspicious pancreatic masses. Pancreatic adenocarcinomas (PDACs) usually have a dense fibroblastic stroma and typically appear as a hypoattenuating lesion to the pancreatic parenchyma on the venous phase; however, PDACs may exhibit isoattenuation, hindering detection, as the lesion is difficult to differentiate from pancreatic parenchyma. Additionally, staging can be overestimated or misdiagnosis can occur due to pancreatitis, which can be a complication after an endoscopic biopsy or biliary decompression or can appear along with adenocarcinoma in patients with chronic pancreatitis. As a result, investigators have increasingly explored AI models to segment the pancreas and characterize lesions on cross-sectional imaging to assist with diagnostic and staging accuracy. Gai et al. developed a comprehensive suite of AI algorithms utilizing radiomic-based schemes for pancreatic segmentation, the radiomic feature computation of pancreatic lesions, and the classification as benign or malignant. Their algorithm yielded a pixel-wise accuracy in segmentation of 74% along with an AUC of 0.75 for tumor classification. Notably, this study only evaluated 77 patients, a relatively small dataset for the development of AI algorithms, and has not been validated in a larger population [17].

To aid in the differentiation of autoimmune pancreatitis (AIP) from PC, Ziegelmayer et al. developed a DL and radiomic-based model showing an AUC of 0.9 and 0.8, respectively [20]. Interestingly, fusion models incorporating DL and radiomic features consistently outperform pure radiomic models [14]. Radiomic feature extraction relies on pixel-level annotation and computational analysis. For PC, this is extremely difficult given the vague boundaries often seen on cross-sectional imaging. Furthermore, traditional radiomic techniques only analyze a region of interest (ROI), ignoring the surroundings of the tumor. These areas are effectively captured by DL models which produce more complex, high-level features. Zhang et al. showed a weak linear relationship between radiomic and DL features when used to predict overall survival in PC, suggesting features from radiomic analysis and DL may be complementary to each other, and fusion methods may be able to capture an optimum set of features [19]. In cancers with more defined borders such as gastric cancer and colorectal cancer, radiomic models alone have demonstrated efficacy; however, with PC, fusion models incorporating radiomics and DL have significantly improved model accuracy [19,20].

While multiphasic, cross-sectional imaging with CT remains the gold standard, transabdominal ultrasounds can also play a role in the diagnosis and staging of PC, especially in low-resource settings; however, detailed information regarding resectability may be limited due to the body habitus and clinical experience of the user. One study utilized DL to differentiate chronic pancreatitis alone and chronic pancreatitis-associated PC in contrast-enhanced ultrasonography (CEUS). Using a ResNet50 architecture, Tong et al. attained a sensitivity of 93% and a specificity of 84% across multiple validation cohorts. Furthermore, model performance was slightly superior or comparable with an expert radiologist. In a two-round reader study format, this study demonstrated improved sensitivity and specificity in nearly all radiologists when utilizing the model to aid in diagnosis [16].

Cross-sectional imaging evaluated with AI can also provide information regarding preoperative lymph node (LN) status. An et al. utilized a fused clinical, radiomic, and DL-based model to predict LN metastatic burden with an AUC of 0.92 [18]. This study shows the efficacy of utilizing AI models which, in specific investigative situations, may outperform radiologists in predictions [15]. Using a more objective fusion model to identify LN metastasis can aid in alleviating the variation seen in radiologist-dependent predictions [9], leading to a more standardized preoperative staging schema.

To pathologically confirm PC, a biopsy is typically obtained via EUS-guided fine-needle aspiration (FNA). EUS images pose another avenue for developers to engineer AI algorithms for the classification of PC. Gu et al. designed a retrospective study to develop a DL radiomic (DLR) model utilizing images from EUS and comparing PC classification outcomes with both junior and expert endoscopists. The DLR model had an AUC, sensitivity, and specificity of 0.94, 83%, and 90%, respectively. With the assistance of DLR, the endoscopist outcomes improved in the second phase of the study. Interestingly, with the assistance of DLR, junior endoscopists performed at the level of expert endoscopists when evaluating PC classification accuracy. This study highlights the role of utilizing AI in clinical practice to augment clinical acumen and improve clinician performance [11].

In addition to aiding in the diagnosis of PC through imaging, AI can help develop novel biomarkers for PC detection. Blyuss et al. retrospectively evaluated four novel urinary biomarkers, LYVE1, REG1B, REG1A, and TFF1, in conjunction with serum CA 19-9 to more accurately risk-stratify PC. Comparing multiple models, including ML and DL algorithms, the authors failed to demonstrate the superiority of any specific algorithm but noted the excellent performance of all algorithms. The chosen logistical regression model utilizing the urinary biomarkers demonstrated an AUC of 0.94, improving to an AUC of 0.96 with the addition of CA 19-9 [33].

Similarly, Al-Fatlawi et al. demonstrated that in conjunction with Ca 19-9, RNA-based variants can be utilized to differentiate between resectable PC, non-resectable PC, and chronic pancreatitis through DL with an AUC of 0.9. Additionally, their neural network was able to identify two mutations, *B4GALT5* and *GSDMD*, which are closely related to PC progression and survival [26]. With the addition of genomic variants, not only can we diagnose PC at an earlier stage, but we can potentially offer patients a more personalized treatment regimen and prognostication as novel targeted therapeutic options are developed.

PACpAInt, a histology-based DL model developed by Saillard et al., utilizes histological slides to differentiate subtypes of PC—classical/basal and stromal active/inactive—which are historically only differentiable through RNA sequencing, a costly technique [25]. PACpAInt correctly predicts PC subtypes at the whole slide level from both biopsies and surgical specimens as well as disease-free and overall survival. The value of knowing the pathologic subtype can have a profound impact on management. For example, the basal type has a poorer prognosis linked to early metastasis and FOLFIRINOX resistance compared to the classical subtype [34]. The potential of AI, specifically DL, to aid in the development of novel biomarkers for early detection is promising and may aid in improving outcomes for patients with PC.

#### 3.1.2. Applications in Predicting Postoperative Outcomes

While AI applications for cross-sectional imaging and ultrasounds have focused on the diagnosis, segmentation, and prediction of PC staging, AI models can also predict intraoperative or postoperative outcomes. Surgical treatment for PC often involves either a pancreatoduodenectomy (PD) or a distal pancreatectomy with an adequate lymphadenectomy. Complication rates can be as high as 50%, even in high-volume centers. The application of AI may aid clinicians in predicting outcomes to optimize perioperative management for patients with PC.

Postoperative pancreatic fistula (POPF) is one of the most clinically relevant complications following pancreatectomy. The development of POPF is highly correlated with the texture of the pancreas along with the size of the pancreatic duct. The gland texture and duct size are often difficult to accurately evaluate preoperatively on cross-sectional imaging, making these variables targets of investigational studies with AI algorithms. To predict POPF, Kambakamba et al. utilized ML to calculate texture-based features such as histologic fibrosis, histologic lipomatosis, and intraoperative hardness for algorithm development. Their algorithm had an AUC, sensitivity, and specificity of 0.95, 96%, and 98%, respectively. This algorithm significantly outperformed available clinical risk scores, which typically have an AUC of 0.7–0.8 [12].

Intraoperative outcomes can significantly impact patient survival in PC. For example, the resection margin status has a significant impact on overall survival. Additionally, the preoperative prediction of an R0 resection, a microscopically negative margin, can potentially impact a clinician’s decision to administer neoadjuvant therapy versus upfront surgery. Chang et al. utilized a 3D convolutional neural network (CNN) to predict the postoperative margin status with an 81% accuracy based on preoperative cross-sectional imaging [13]. The applications of AI in diagnostics and prognostication may significantly impact clinical decision making with the potential to improve survival for patients with PC.

#### 3.1.3. Applications in Treatment Response

Over the past decade, a paradigm shift has occurred regarding the role of neoadjuvant therapy in PC. Treatment strategies now include neoadjuvant systemic therapy rather than up-front surgical resection for many patients with PC. Multiagent chemotherapeutic regiments such as FOLFIRINOX (leucovorin, 5-fluorouracil, irinotecan, and oxaliplatin) and gemcitabine/nab-paclitaxel significantly improve overall survival [35,36]. It is important to monitor and, if possible, predict responses to chemotherapy, as clinical improvement or disease progression will impact management decisions [37]. CA 19-9 is used as a biomarker to monitor responses to neoadjuvant chemotherapy; however, there are many limitations with this approach. First, nearly 10% of the population does not possess the Lewis antigen and, therefore, will never have an elevation in CA 19-9. In addition, other clinically relevant factors may confound CA 19-9 results (i.e., falsely high values with hyperbilirubinemia). Finally, while a reduction or normalization in CA 19-9 is associated with improved survival, the association with a histopathologic response is inconsistent [38].

AI approaches may enhance the prognostic accuracy of current biomarkers used to monitor neoadjuvant therapy responses. Watson et al. utilized preoperative cross-sectional imaging and a >10% decrease in serum CA 19-9 after neoadjuvant treatment to predict histopathologic responses. A CNN using the LeNet architecture, a 5-layer CNN, was created to predict pathologic responses. In isolation, preoperative imaging yielded an AUC of 0.74, and using CA 19-9 decreased by >10% yielded an AUC of 0.56. The CNN fusion model including both biomarkers yielded an AUC of 0.79 (*p* < 0.001). As this was a pilot study, only 81 patients were included, which is a small group of patients to train and validate a DL algorithm [39]. The utilization of DL enables the use of traditional biomarkers in novel ways.

#### 3.1.4. Applications of Large Language Models

In November of 2022, ChatGPT-4 was released to the public and took the medical community by storm. GPT-4 is a transformer-based large language model (LLM) that utilizes an extensive corpus of text data from the internet to predict a series of words to formulate a response to any question it is asked. This includes having provided responses at or near a threshold of passing the United States Medical Licensing Exam (USMLE). Due to its novelty, the applications and validity of GPT-4 are continuously explored in the medical field. Walker et al. aimed to assess the reliability of ChatGPT specifically in the management of five common hepatobiliary and pancreatic conditions: gallstone disease, pancreatitis, liver cirrhosis, pancreatic cancer, and hepatocellular carcinoma. Agreements between guideline recommendations and ChatGPT responses were 60% with a consistency of 100% [22]. Upon a detailed analysis, this study found that while an AI-based chatbot may not be able to provide an accurate recommendation based on medical guidelines, ChatGPT can provide information superior to that found on the internet, and although the application of LLMs in healthcare is still in its infancy, these models may offer significant advantages in information collection and delivery in the future.

In addition to chatbots, LLMs and natural language processing (NLP) can also mine information from text-based databases. Do et al. utilized NLP to accurately classify radiology reports with metastatic diseases and consequently demonstrated patterns of metastatic disease for prevalent cancers such as prostate, breast, pancreatic, and colorectal cancer [21]. They were able to demonstrate different metastatic disease patterns, such as the spread of breast and prostate cancer to the bones compared to liver metastasis in colorectal and pancreatic cancer, with the use of NLP on structured radiology reports. Additionally, this study also resulted in the creation of a large database of over 90,000 patients with labeled radiology reports which can then be utilized to train robust AI algorithms in other domains of healthcare.

#### 3.1.5. Applications of Generative Adversarial Networks

Generative AI is currently captivating interest from many sectors globally. Examples of generative AI applications include ChatGPT and Bert for text-based applications and DALL-E for an image-based generator. The potential applications of generative AI, specifically image-based techniques, in the care of patients with PC are extensive and just starting to be explored. Unfortunately, the engineering of generative applications in the healthcare industry is difficult due to the lack of large databases and the commonly unsupervised nature of generative AI. However, generative adversarial networks (GANs) propose a clever solution to apply generative AI in a supervised fashion.

GANs consist of two modules: a generator and a discriminator. The generator, which is trained on a known corpus of data, produces an image or a string of text, and the discriminator then attempts to classify the image or text as “real” (from the dataset) or “fake” (generated). Both models are trained together in an adversarial manner until the discriminator is fooled more than half of the time. At this point, the generator has learned the ability to generate images or text indistinguishable as real or fake (Figure 4).

The applications of GANs in cross-sectional imaging can have a powerful impact on the management of PC. For example, Hooshangnejad et al. created deepPERFECT, a GAN-based DL model to generate CT scans with an adequate image quality to assess and evaluate the administration of radiation therapy (RT). During the process of evaluating a patient with PC for RT, the patient has already obtained baseline cross-sectional imaging. However, due to the differences in patient setup and image acquisition techniques, the initial imaging is often not useful when planning RT, so another “planning” imaging study is obtained. Obtaining planning imaging has been shown to create delays in the administration of RT. deepPERFECT can evaluate diagnostic CT scans and calibrate the differences in image acquisition and patient setup to synthesize planning CT scans with a dice similarity coefficient (DSC) of 0.93 [24]. By implementing deepPERFECT, patient wait times before the start of treatment can be reduced by one week along with a reduced cost burden to the patient and hospital system. Similarly, Momin et al. utilized GANs to predict SBRT dose distributions and aid in identifying organs at risk (OAR). Their group evaluated different adversarial and non-adversarial frameworks to predict SBRT dose distributions from cross-sectional imaging with an accuracy of 91%. No significant differences were noted when compared to the ground truth labeled by expert radiation oncologists (*p* < 0.05) [23].

### 3.2. Pancreatic Cystic Lesions

IPMN and MCN are mucinous PCLs that can develop PC and account for approximately 8% of all PCs [4]. Timely surgical resection offers the only opportunity for curative treatment. Definitive diagnosis can involve invasive procedures such as through-the-needle biopsies (TTNBs) or confocal laser endoscopy (CLE) [41]. Pancreatectomy in patients with high-grade dysplasia present in the cyst, a precursor state, as opposed to even early invasive cancer, has a substantial impact on the 5-year survival. Pancreatectomy in patients with a PCL containing only low-grade dysplasia, not an obligate precursor lesion, unnecessarily exposes patients to significant surgical risks without any impact on outcomes. Current clinical guidelines on operative timing insufficiently predict high-grade dysplasia or early invasive carcinoma. Early studies evaluating the diagnostic accuracy of cross-sectional images before the utilization of DL revealed accuracies of 40–45% [27]. However, with the utilization of AI, Liang et al. evaluated CT data and developed both radiomic and fused DLR models showing an AUC of 0.97 in the diagnosis of IPMN and MCN with an AUC of 0.92 for the diagnosis of serous cystadenoma (SCA) [30]. The fusion DLR models outperformed the pure radiomic models once again.

IPMNs represent approximately 50% of all incidentally detected PCLs [42]. They arise from the ductal epithelium of the main pancreatic duct or side branches of the main pancreatic duct. The incidence of invasive carcinoma found in resected IPMNs varies significantly based on several clinical factors. An analysis of over 100 institutions in the United States found an invasive carcinoma incidence of 23% in resected IPMNs [43]. In most patients, the oncogenic process takes many years, offering an opportunity to intervene early and prevent the development of invasive diseases, greatly impacting patient outcomes. However, due to the significant morbidity associated with a pancreatectomy, the decision to perform surgery in patients with an IPMN must be weighed against the risk of high-grade dysplasia or carcinoma to avoid unnecessary surgery in patients with only low-grade dysplasia or no dysplasia. Risk assessment is critical to the management of patients with IPMN. Kuwahara et al. utilized common CNN architectures, such as AlexNet and ResNet, to predict malignancy in an IPMN with an AUC, accuracy, sensitivity, and specificity of 0.91, 94%, 96%, and 93%, respectively [28]. When compared to the 2006 Sendai guidelines (sensitivity 100%; specificity 7.6%) and the 2012 Fukuoka guidelines (sensitivity 84.8%; specificity 45%) [44], the utilization of AI to risk-stratify patients with IPMNs offered significant advantages and may prevent patients from unnecessary pancreatectomy [29].

### 3.3. Pancreatic Neuroendocrine Tumors

Pancreatic neuroendocrine tumors (PNETs) arise from the endocrine cells of the pancreas and are either functioning or non-functioning tumors, the latter of which have a higher prevalence of malignancy. Complete surgical resection is often curative for non-metastatic PNETs. The postoperative recurrence and rate of metastasis vary by subtype. AI can be used as an adjunct to predict postoperative recurrences. Song et al. utilized preoperative clinical features in conjunction with a DLR model to predict recurrences. They attained a maximum AUC of 0.7 in the validation cohort including preoperative CT scans in the arterial phase, as PNETs are characteristically hypervascular [31]. Unfortunately, this algorithm was trained on a small sample size of 56 patients in the training cohort and 18 patients in the validation cohort. This is largely due to the rarity of PNETs and the paucity of large databases.

As previously described, GANs can produce artificial images and have been utilized in generating CT scans to plan RT and predict dose distributions during the administration of SBRT to treat PC. GANs can also be leveraged when there is a lack of large databases with rare diseases such as PNETs. Gao et al. utilized GANs to generate synthetic images of PNETs to augment a small dataset of 96 patients and develop a DL algorithm to predict the World Health Organization (WHO) grade of a PNET with an accuracy of 85% and an AUC of 0.91 [32].

### 3.4. Ethical Considerations

While the integration of AI in healthcare generates tremendous excitement, it also poses ethical issues that must be addressed before wider adoption. The WHO has identified six core principles that should be adhered to when designing, developing, and deploying AI for health: “(1) protect autonomy; (2) promote human well-being, human safety, and the public interest; (3) ensure transparency, explainability, and intelligibility; (4) foster responsibility and accountability; (5) ensure inclusiveness and equity; (6) promote AI that is responsive and sustainable” [45].

Unfortunately, the complexity of modern AI systems developed by engineers and computer scientists makes interpretability and meaningful scrutiny difficult for untrained clinicians. The opacity created by AI systems when producing a recommendation must be mitigated such that clinical users, many with minimal technical backgrounds, can understand the output of AI systems and communicate the findings to their patients. Furthermore, if a clinician cannot comprehend the output of an AI system, it may be challenging to justify their action if they choose the generated recommendation. The increasing utilization of AI in high-risk clinical situations demands accountable, equitable, and transparent AI designs [46].

The use of AI to assist in clinical decision making has also changed the traditional healthcare paradigm by creating a new stakeholder dynamic. Clinicians are utilizing AI systems on the front line and thus have a stake in the safe introduction of new technologies to the clinical setting. Interestingly, clinicians, unlike engineers and technologists, are legally accountable for their actions. Technologists, on the other hand, abide by a set of ethical principles of practice. This difference accounts for the current dispute in the accountability for errors in AI systems. For a safe and effective roll-out of innovative AI technologies, governing entities composed of multiple stakeholders, such as physicians and technologists, must be created to ensure accountability and establish legislation regarding the utilization of AI in the healthcare industry. Responsible oversight can also support equitable and inclusive AI algorithms. Furthermore, given the large datasets required to generate clinically robust and inclusive algorithms, data access agreements and cybersecurity considerations, in accordance with institutional and governmental patient privacy laws, are foundational components to the safe development and adoption of AI technology. Clinicians and technologists must work side by side to ensure the six core principles listed by the WHO are followed while implementing AI systems in clinical practice.

## 4. Conclusions

The applications of AI are limitless in the field of medicine. PC poses a significant challenge given the difficulty with early diagnosis and predilection for early lymph node metastasis leading to a poor 5-year overall survival rate. The integration of DL and AI applications can aid in all facets of care for patients with PC: earlier diagnosis, improving biomarker accuracy, monitoring treatment response, predicting perioperative and survival outcomes, and optimizing resource utilization. With exponentially increasing applications of AI in the care of cancer patients, governing boards will need to be proactive in the oversight and regulation of predictive algorithms, advocating for transparent and ethical algorithm developments. With the addition of AI to the armamentarium clinicians can use to treat patients with PC, we will hopefully see an improvement in outcomes for patients with PC.

## Figures and Tables

**Figure 1 cancers-16-00436-f001:**
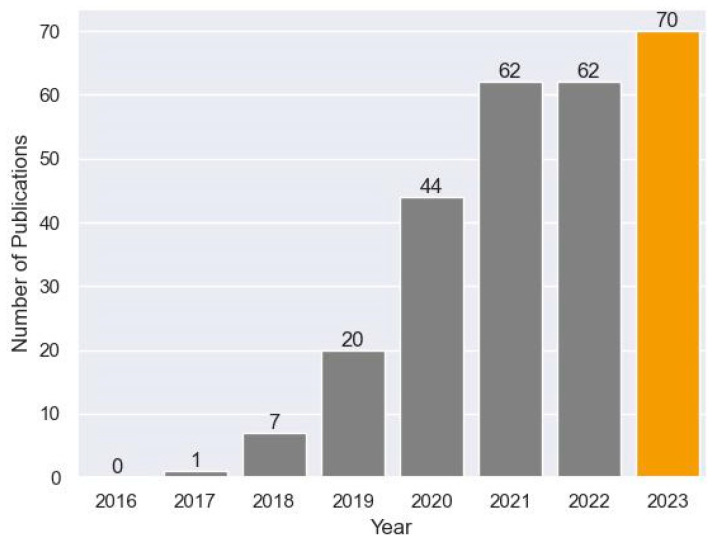
Number of publications in a PubMed search using deep learning and pancreas as key terms from 2016–2023.

**Figure 2 cancers-16-00436-f002:**
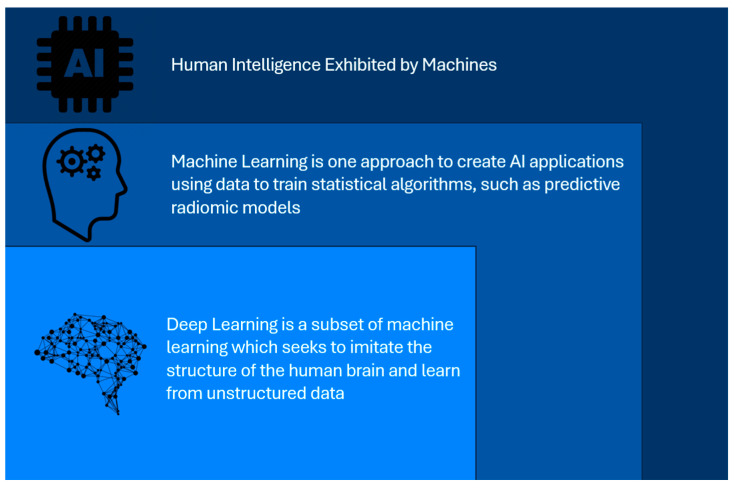
Relationship of AI, ML, and DL. Neural networks are a type of DL. Radiomics are a subset of ML algorithms [8].

**Figure 3 cancers-16-00436-f003:**
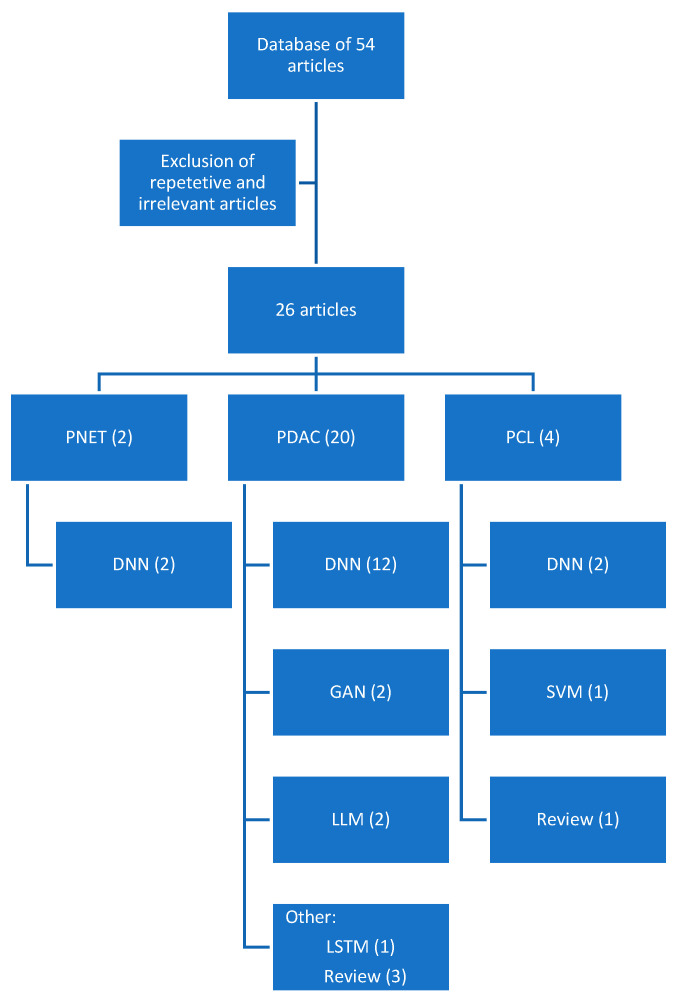
Methodology of review article: deep neural network (DNN); generative adversarial network (GAN); large language model (LLM); support vector machine (SVM).

**Figure 4 cancers-16-00436-f004:**
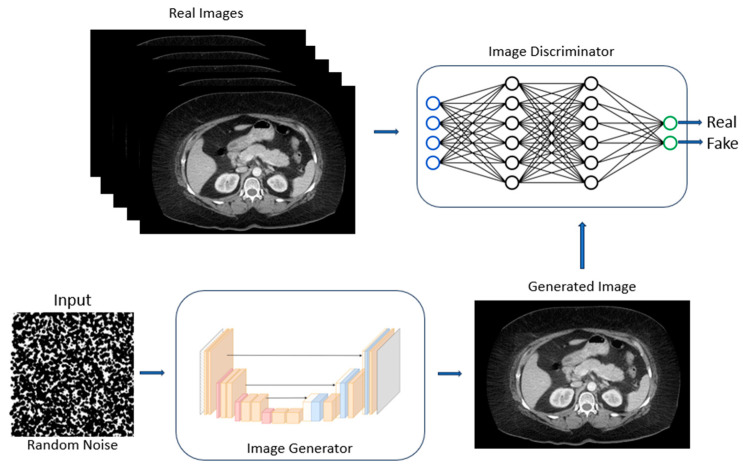
An example of a GAN showing the adversarial nature of generated images being scored by the discriminator as “fake” or “real” until the discriminator has been fooled, in essence creating fake images which can plausibly be considered as real [40].

**Table 1 cancers-16-00436-t001:** Summary of articles utilized in this review.

Author	Title	Article Type	Pathology	Algorithm	Performance (AUC-ROC)
Fu et al. [9]	A deep-learning radiomics based lymph node metastasis predictive model for pancreatic cancer: A diagnostic study	Retrospective	PDAC	CNN	0.85
Faur et al. [10]	Artificial intelligence as a noninvasive tool for pancreatic cancer prediction and diagnosis	Review article	PDAC	n/a	n/a
Gu et al. [11]	Prospective assessment of pancreatic ductal adenocarcinoma diagnosis from endoscopic ultrasonography images with the assistance of deep learning	Prospective	PDAC	CNN ResNet50	0.9
Kambakamba et al. [12]	The potential of machine learning to predict postoperative pancreatic fistula based on preoperative, non-contrast-enhanced CT: A proof-of-principle study	Retrospective	Postoperative Whipple	Multiple	0.78
Chang et al. [13]	Machine-learning based investigation of prognostic indicators for oncological outcome of pancreatic ductal adenocarcinoma	Retrospective	PDAC	CNN	0.79/0.85 (LN/margin status)
Wei et al. [14]	A multidomain fusion model of radiomics and deep learning to discriminate between PDAC and AIP based on 18F-FDG PET/CT images	Retrospective	PDAC	CNN VGG11	0.96
Bian et al. [15]	Artificial Intelligence to Predict Lymph Node Metastasis at CT in Pancreatic Ductal Adenocarcinoma	Retrospective	PDAC	CNN	0.91
Tong et al. [16]	Deep learning radiomics based on contrast enhanced ultrasound images for assisted diagnosis of pancreatic ductal adenocarcinoma and chronic pancreatitis	Retrospective	PDAC	CNN ResNet50	0.9
Gai et al. [17]	Applying a radiomics-based CAD scheme to classify between malignant and benign pancreatic tumors using CT images	Retrospective	PDAC	LSTM	0.7
An et al. [18]	Deep learning radiomics of dual-energy computed tomography for predicting lymph node metastases of pancreatic ductal adenocarcinoma	Retrospective	PDAC	CNN ResNet18	0.87
Zhang et al. [19]	Improving prognostic performance in resectable pancreatic ductal adenocarcinoma using radiomics and deep learning features fusion in CT images	Retrospective	PDAC	CNN	0.84
Zeigelmayer et al. [20]	Deep Convolutional Neural Network-Assisted Feature Extraction for Diagnostic Discrimination and Feature Visualization in Pancreatic Ductal Adenocarcinoma (PDAC) versus Autoimmune Pancreatitis (AIP)	Retrospective	PDAC	CNN VGG19	0.9
Do et al. [21]	Patterns of Metastatic Disease in Patients with Cancer Derived from Natural Language Processing of Structured CT Radiology Reports over a 10-year Period	Retrospective	PDAC	NLP	Accuracy 90%
Walker et al. [22]	Reliability of Medical Information Provided by ChatGPT: Assessment Against Clinical Guidelines and Patient Information Quality Instrument	Retrospective	PDAC	GPT 4	60% Agreement
Momin et al. [23]	Learning-based dose prediction for pancreatic stereotactic body radiation therapy using dual pyramid adversarial network	Retrospective	PDAC	GAN	Accuracy 87%
Hooshangnejad et al. [24]	deepPERFECT: Novel Deep Learning CT Synthesis Method for Expeditious Pancreatic Cancer Radiotherapy	Prospective	PDAC	3D UNET GAN	DSC 93%
Bonmati et al. [5]	Pancreatic cancer, radiomics and artificial intelligence	Review Article	PDAC	n/a	n/a
Saillard et al. [25]	Pacpaint: a histology-based deep learning model uncovers the extensive intratumor molecular heterogeneity of pancreatic adenocarcinoma	Retrospective	PDAC	ANN	0.8–0.9
Al-Fatlawi et al. [26]	Deep Learning Improves Pancreatic Cancer Diagnosis Using RNA-Based Variants	Retrospective	PDAC	DNN	0.96
Huang et al. [27]	Pancreatic Cystic Lesions: Next Generation of Radiologic Assessment	Review Article	PCL	n/a	n/a
Kuwahara et al. [28]	Usefulness of Deep Learning Analysis for the Diagnosis of Malignancy in Intraductal Papillary Mucinous Neoplasms of the Pancreas	Retrospective	PCL	CNN ResNet50	0.9
Chakraborty et al. [29]	CT radiomics to predict high-risk intraductal papillary mucinous neoplasms of the pancreas	Retrospective	PCL	SVM/RF	0.8
Liang et al. [30]	Classification prediction of pancreatic cystic neoplasms based on radiomics deep learning models	Retrospective	PCL	ANN	0.9
Song et al. [31]	Predicting the recurrence risk of pancreatic neuroendocrine neoplasms after radical resection using deep learning radiomics with preoperative computed tomography images	Retrospective	PNET	CNN UNET	0.8
Gao et al. [32]	Deep learning for World Health Organization grades of pancreatic neuroendocrine tumors on contrast-enhanced magnetic resonance images: a preliminary study	Retrospective	PNET	CNN GAN	0.9

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
