# Peer review of "Deep Learning Applications in Pancreatic Cancer"

_cancers, 2024, doi:10.3390/cancers16020436_

Round 1

Reviewer 1 Report

Comments and Suggestions for Authors

Very interesting and well written review. Maybe a table summarizing the ongoing and past studies in the field would be useful to the reader.

The authors should comment that the current standard of care for the diagnosis of PDAC in non-surgical patients is EUS-FNB (not FNA). (in this regard cite the recent paper: PMID: 31031330)

THe authors should mention the main diagnostic tools for pancreatic cystic lesions, such as CLE and TTNB (in this regard cite the recent series PMID: 35451041)

Author Response

Dear Editor,

Thank you for your insights. We really appreciate your feedback. Here are the changes made based on your response:

  1. Added a table to page 4-5 highlighting all studies used in the review
  2. Changed FNA to FNB (line 43)
  3. Added a sentence regarding diagnosis incorporating TTNB and CLE with added source (line 311-312)

Thank you again for your response.

Sincerely,

Hardik Patel

Reviewer 2 Report

Comments and Suggestions for Authors

The review offers a comprehensive overview of the current landscape of AI applications in the field of pancreatic cancer. However, I have a few observations regarding the content:

  1. 1. The authors extensively explore the use of AI in pancreatic cancer, specifically addressing pancreatic ductal adenocarcinoma, pancreatic cystic lesions, and pancreatic neuroendocrine tumors. Despite this, there appears to be a discrepancy between the classification presented in the subtitle of the results section and that depicted in Figure 3. To enhance clarity, I recommend the inclusion of a table organizing the articles discussed in the review.

  2. 2. Could you provide clarification on the meaning of "other" in Figure 3?

  3. 3. Historically, AI research has predominantly focused on medical imaging. In the context of pancreatic cancer, CT scans remain the primary diagnostic, management, and monitoring tool. Conversely, natural language processing or large language models represent an alternative approach to AI application in medicine. Are there additional applications for pancreatic cancer, such as risk stratification, histology classification, gene analysis, or any other relevant areas? Further exploration of these aspects would enrich the discussion.

Author Response

Dear Editor,

Thank you for your insight on our article. We really appreciate your time and considerations. Here are the changes made based on your response:

  1. A table added to page 4-5 highlighting all studies incorporated in our review
  2. In itemized list highlighting what "other" means in Figure 3
  3. Added two paragraphs regarding application of deep learning to histopathology and RNA sequencing with regards to pancreatic cancer. (Line 171 - 189)

Thank you so much for helping us truly elevate the content of our article.

Sincerely,

Hardik Patel

Round 2

Reviewer 1 Report

Comments and Suggestions for Authors

The revised version of the manuscript is OK. Thank you!

Reviewer 2 Report

Comments and Suggestions for Authors

I have reviewed the revisions made to your article and commend your efforts. The addition of a comprehensive table, clarification in Figure 3, and the expansion on deep learning applications demonstrate a commitment to enhancing the manuscript.

Well done on addressing the feedback, and I look forward to the continued success of your work.